# Physico-Chemical Properties of Sugar Beet Pectin-Sodium Caseinate Conjugates via Different Interaction Mechanisms

**DOI:** 10.3390/foods8060192

**Published:** 2019-06-03

**Authors:** Juyang Zhang, Bettina Wolf

**Affiliations:** Division of Food Sciences, School of Biosciences, University of Nottingham, Sutton Bonington Campus, Loughborough LE12 5RD, UK; B.Wolf@bham.ac.uk

**Keywords:** sodium caseinate, sugar beet pectin, electrostatic interaction, enzymatic cross-linking, Maillard reaction, polysaccharide-protein complex, acidic pH and thermal processing

## Abstract

Polysaccharides and proteins are frequently conjugated through electrostatic attraction, enzymatic cross-linking, and heat treatment (Maillard reaction) to obtain food structuring ingredients, mostly for their application as emulsifiers. The conjugate partners and their interaction type affect performance at acidic or neutral pH and during thermal processing, thus requiring careful selection. Here, the aggregate properties (particle size, conjugate charge, shear viscosity) of three types of sugar beet pectin (SBP)-sodium caseinate (SC) 1:1 conjugates, at acidic and neutral pH (4.5; 7), as well as their thermal processing stability (80 °C), were investigated. The enzymatically cross-linked SBP:SC was more acid tolerant than the electrostatically interacting conjugates. Maillard cross-linked conjugates aggregated at pH 4.5, suggesting poor emulsifier performance in acidic conditions. At pH 7, the three conjugate types showed similar aggregate properties. The results are discussed in terms of structural re-arrangement.

## 1. Introduction

The quality of a food product is characterised by its texture, structure, and stability as imparted by the intrinsic functional properties of the ingredients and their interactions during the manufacturing process. Polysaccharides and proteins both improve the microstructure and stability of multiphase food systems because of their physicochemical functional properties at interfaces [1]. The structural functionality of a polysaccharide and protein concomitantly present in a formulation may be the result of the interaction with each other, prompting an improvement in individual biopolymers’ functional properties [2]. Indeed, polysaccharide-protein complexes are widely applied in the field of encapsulation, protection and delivery of functional food ingredients, such as bioactive lipids, minerals, enzymes, peptides and so forth [3,4]. Moreover, the application of polysaccharide-protein complexes as fat replacer has been reported. Examples include, complexes prepared from milk protein and xanthan gum [5], milk protein and carrageenan [6], soy protein and xanthan gum [7], as well as casein and pectin [8]. Milk proteins, such as casein, are aggregated with lowering pH conditions. To improve the solubility of casein under acidic conditions, some polysaccharides have been introduced, for instance, pectin [9,10].

Casein presence is essential for the structural integrity of dairy products, such as cream, cheese and butter, and also provides essential amino acids. In addition, the use of casein micelles has been suggested as an ideal encapsulation vehicle for nutraceuticals, such as fat-soluble vitamins [11,12]. There are four types of casein: α_s1_-casein, α_s2_-casein, β-casein, and κ-casein, all of which possess different structures and functionality. All four caseins in milk aggregate into casein micelles spontaneously as a result of the interaction with calcium phosphate [11,13,14,15]. Both αs1-casein and β-casein are major caseins, which have the tendency to self-associate because of their amphiphilic nature. However, β-casein forms spherical micelles, whereas α_s1_-casein is characterised by chain-like aggregates, due to the presence of a stronger net negative charge, as both proteins are at neutral pH [13]. In contrast, κ-casein is a glycoprotein, which can sterically stabilise casein micelles formed by α_s1_-casein, α_s2_-casein and β-casein [13,15]. Pectin is a heteropolysaccharide, which is extensively used in food production, due to its gelling and stabilising properties [16]. The structure of pectin consists of a backbone of 1-4-linked d-galacturonic acid (GalA) units interrupted by some rhamnogalacturonan segments, which combine galacturonic acid residues and a-l-rhamnopyranose. Side chains are glycosidically linked to O-4 and/or O-3 of l-rhamnopyranose, and O-2 or O-3 of some of the galacturonosyl residues. In addition, there are sugar constituents attached as side chains, such as d-galactose and l-arabinose. These sugars are present in galactan, arabinogalactan and arabinan. In addition, ferulic acid, which has a phenolic acid structure, ester-links to either, arabinose or galactose [16,17,18,19]. Generally, pectin is extracted from plant cell walls, such as citrus peels, apple pomace and sugar beet pulp. Sugar beet pectin can be obtained from sugar beet pulp during the extraction of sugar [17].

The study, presented here, was dedicated to evaluating the physicochemical properties of polysaccharide-protein conjugates, that were prepared from sugar beet pectin (SBP) and sodium caseinate (SC), for the future application of these conjugates as emulsifiers. In principle, there are three methods applicable for the preparation of food-grade polysaccharide-protein conjugates: Electrostatic interaction, enzymatic methods and Maillard reaction. The most popular method for the formation of polysaccharide-protein conjugates is electrostatic interaction between opposing charges on the two biopolymers [20]. Polysaccharides are characterised by anionic, cationic or non-ionic charges, which correlate with the nature of functional groups, but are also influenced by pH conditions, based on the pKa value of the ionisable side groups [21]. SBP possesses only carboxylates (-CO^2−^, pKa about 2.5 to 4.5) as ionisable group and is net-negatively charge [22]. SC is a protein that possesses both positive and negative charges, because of protonated amino side groups (-NH^3+^) at a pH of below 10 and deprotonated carboxylate side groups (-CO^2−^), at a pH higher than two, respectively. The isoelectric point (pI), the pH at which the net-charge is zero, is ~4.6 [21].

Unlike the electrostatic interaction method, an enzymatic method is based on chemical cross-linking. Laccase is a multi-copper, polyphenol oxidase obtained from bacteria, fungi and plants [23,24]. It generates free radicals by oxidising various compounds, such as amines, thiols and iodine [25], typically phenoxy radicals with a loss of single electrons to form radicals, including quinones and/or phenoxy radicals [24,26]. In addition, laccase is capable of catalysing ferulic acid, which is a phenol structure esterified to the arabinose side-chain at the backbone of the rhamongalacturonan I side chains in SBP. In the presence of oxygen these side chains are enzymatically oxidised into FA dihydrodimers (diFAs) [27]. It has previously been reported that the emulsification properties of SBP improved following inter-molecular cross-linking via laccase catalysis [26]. Furthermore, SBP has been shown to improve the stabilisation of emulsion systems prepared by using whey protein [27,28,29], fish gelatine [30,31] and β-lactoglobulin [32,33] via cross-linking of SBP through the addition of laccase. In addition, laccase has the ability to oxidise tyrosine and tyrosine-containing peptides, as ferulic acid can become covalently cross-linked into the polymer structure through an ether bond [23]. Laccase can also oxidise amino acids, such as cysteine and phenol-based tryptophan [25,29,34]. It has also been reported that laccase can catalyse SC and can form inter-molecular bonds between the caseinate molecules [35]. Therefore, it can be inferred that three structures may be involved in SBP-SC conjugates via laccase catalysis, namely SBP-SC, SBP-SBP and SC-SC conjugates.

Finally, the Maillard reaction is a non-enzymatic browning reaction that occurs during heating, roasting, baking and frying in the presence of both carbohydrates and proteins in food products [36]. The Maillard reaction has favourable effects, such as colour and flavour formation during roasting, baking or frying, whereas it is unfavourable in processes, such as drying, pasteurisation and sterilisation [37]. The Maillard reaction is a chemical reaction, cross-linking aldehydes and amines, through a well-established oxidation-reduction pathway [21]. For common proteins and polysaccharides, the Maillard reaction occurs between amino compounds and reducing sugars as reactants [37]. The major variables, that impact upon the Maillard reaction, are not only temperature, time and relative humidity, but also the nature and proportion of each polymer [38]. Generally, the Maillard reaction involves three stages, and the first stage may be enough for improving the emulsification properties of conjugates [39]. This stage naturally occurs between the carbonyl group of polysaccharides and amino acids of proteins, in order to isomerise the Amadori product, which is the product of the condensation process, via the formation of a Schiff base with the release of water and the Amadori rearrangement [20,36,39].

All three types of conjugates were created from SBP and SC and their aggregation properties evaluated by particle size, Zeta (ζ)-potential and dynamic viscosity. Relevant to application in foods, the effect of environmental stress factors, including pH conditions, pH 4.5 and pH 7, and thermal treatment at 80 °C for 10 min, on the aggregation properties was also assessed. Finally, SBP-SC interaction models are proposed.

## 2. Materials and Methods

### 2.1. Materials

The main materials used to prepare the polysaccharide-protein conjugate emulsifiers, and appropriate reference samples, were sugar beet pectin (Herbstreith & Fox KG, Neuenbürg, Germany) sodium caseinate, citric acid monohydrate and sodium citrate dihydrate for the preparation of citrate buffers (pH 4 and pH 5), hydrochloric acid and sodium hydroxide to adjust the pH with 1 M solutions, sodium azide as antimicrobial (purchased from Fisher Scientific, Loughborough, UK), ferulic acid, potassium bromide, syringe filters (0.45 µm, ø 15mm, Whatman GE Healthcare) and laccase enzyme (purchased from Sigma-Aldrich, Gillingham, UK). The laccase activity was reported by the supplier as 0.87 units per mg (AU) of the enzyme. Deionized water (electrical conductivity < 2 µS cm^−1^) produced on-site was used throughout.

### 2.2. Dispersion Preperation

A 0.1 M Citrate buffer (pH 5) was prepared by mixing 20.5 mL of 0.1 M citric acid and 29.5 mL of 0.1 M sodium citrate with 50 mL of water on a magnetic stirrer at 500 rpm and 25 °C for 30 min. The solution was then diluted with water to obtain 50 mM citrate buffer at pH 5 to use as solvent. Dispersions of different ferulic acid concentrations (0.1–1.5 mg/100 g) were prepared in 50 mM citrate buffer (pH 5) by dissolving ferulic acid powder in the appropriate amount of water on a magnetic stirrer at 500 rpm and 25 °C for 2 h to ensure full dissociation.

An amount of 0.1 or 1 *w*/*w* % SBP dispersion was prepared by dissolving 1 g of SBP powder in the appropriate amount of water and citrate buffer (50 mM), followed by mixing on a magnetic stirrer at 500 rpm and 25 °C overnight. SC dispersion, at a total polymer concentration of 1 *w*/*w* %, at pH 7, was prepared by dissolving 1 g of SC powder in the appropriate amount of water on a magnetic stirrer at 500 rpm and 25 °C for one hour to ensure full hydration. A 1:1 SBP:SC dispersion was prepared by mixing 1 *w*/*w* % SC and SBP dispersions, with the appropriate amount of deionized water, containing 0.02 *w*/*w* % sodium azide.

Different laccase/sugar beet pectin ratios were prepared at pH 5, namely: 1.15 mg/4 mg (1 AU); 5.75 mg/4 mg (5AU); 11.5 mg/4 mg (10AU); and 23 mg/4 mg (20 AU). Thus, 2.3 *w*/*w* % enzyme was prepared by dispersing 2.3 g of laccase in the appropriate amount of water and 50 mM citrate buffer, followed by 1 h of stirring for complete hydration. Then, 50 g laccase/SBP mixture dispersion was prepared by mixing 2.3 *w*/*w* % laccase and 0.1 *w*/*w* % SBP dispersion with the appropriate amount of citrate buffer (50 mM) at 25 °C.

### 2.3. Analysis of Ferulic Acid Concentration and Selection of Experimental Conditions for Laccase-Catalysed Cross-Linking of Sugar Beet Pectin and Sodium Caseinate

The proportion of ferulic acid in the SBP, obtained for this study, needed to be quantified to select the appropriate experimental conditions for the laccase-catalysed cross-linking of SBP and SC. A previously published method, based on UV visible spectroscopy [30], was applied. The maximum absorbance for SBP is at 325 nm wavelength, which is attributed to the presence of ferulic acid groups in this material [30]. The absorbance of the different concentrations of ferulic acid dispersions was determined at 325 nm, by using a UV-visible light spectrophotometer (HP 8453 Agilent, Agilent Technologies, Waldbronn, Germany) at 25 °C. A total of 50 mM citrate buffer (pH 5) was used as a blank. Based on the standard curve reported in Appendix A (Appendix A) and absorbance of a 0.1% *w*/*w* SBP dispersion, it was established that the SBP contained 1.33% ferulic acid. Then, a standard curve of the calibration dispersions was plotted as a function of the ferulic acid concentration from 0.1–1.5 mg/100 g. To calibrate the absorbance signal.

Having established the proportion of ferulic acid in the SBP, it was assessed whether laccase would cross-link the SBP molecules following previously published protocol for laccase activity assessment [30,33]. Laccase/SBP mixture dispersions were gently shaken and then immediately sampled into the spectrophotometer and absorbance at 325 nm followed for 60 min. 1, 5 and 10 AU laccase activity were tested and since the absorbance data overlapped, see Appendix A (Appendix A), the intermediate of 5AU was chosen to carry forward.

### 2.4. Preparation of Conjugate Dispersions

Electrostatically, enzymatic and through Maillard cross-linked SBP-SC conjugate dispersions were prepared at a 1:1 by weight mixing ratio as described in the following.

#### 2.4.1. Laccase-Catalysed SBP Dispersions

SBP dispersions were prepared as reference samples. A 0.4 *w*/*w* % laccase-catalysed SBP dispersion was prepared by dissolving 1 *w*/*w* % SBP dispersion in the appropriate amount of water, containing 0.02 *w*/*w* % sodium azide and 10 g 5 AU laccase dispersion on a magnetic stirrer at 500 rpm and 25 °C for 2 h to ensure complete catalysis. The completion of the enzymatic catalysis was determined by visual observation of the colour of the SBP dispersion, changing from turbid to clear brown. The dispersion was stored at 25 °C until further use.

#### 2.4.2. Electrostatically-Stabilised SBP-SC Conjugate Dispersions

To cross-link SBP and SC, the pH of the 1:1 SBP:SC dispersion with a total polymer content of 0.4 *w*/*w* % was adjusted to 4.5 by the addition of either 1 M HCl or 1 M NaOH, as appropriate. This was stirred at 500 rpm and 25 °C for at least 6 h on a magnetic stirrer to ensure complete formation of SBP-SC conjugates. Observation of the completion of the conjugate formation process was possible visually as the SC dispersion was clear, whereas the SBP dispersion was turbid brown; upon conjugate formation the appearance changed from turbid brown to turbid white. The conjugate dispersion was labelled SBP:SC P (P to denote physical cross-linking by electrostatic interaction).

#### 2.4.3. Laccase-Catalysed SBP-SC Conjugates Dispersions

A 1:1 SBP:SC dispersion with a total polymer content of 0.4 *w*/*w* % was prepared by mixing the appropriate amounts of stock dispersions with deionized water containing 5 AU laccase enzyme. After that, the SBP-SC mixture was placed in a magnetic stirrer at 500 rpm and 25 °C for 2 h to ensure complete enzymatic reaction. The initially turbid white and finally turbid light brown appearing dispersion was labelled SBP:SC E (E to denote enzymatic cross-linking by laccase catalysis).

#### 2.4.4. SBP-SC Maillard Conjugates Dispersions

For the preparation of the SBP-SC Maillard conjugates a previously published protocol was followed [36]. SBP and SC were dissolved in deionized water (solid/liquid ratio 1:25) at a ratio of 1:1 while stirring at 700 rpm and 25 °C for 2 h on a magnetic stirrer. After that, the mixture was stored in a freezer at −80 °C for at least 24 h and followed by dehydration in a freeze-drier (Super Modulyo, Edwards, Burgess Hill, UK) at −40 °C and pressure between 7 × 10^−2^ bar and 2 × 10^−2^ bar. The dried solids were placed in a desiccator containing a saturated KBr dispersion and incubated for 48 h in a cabinet (Sanyo/Gallenkamp cabinet, model CF4) at 60 °C, with a relative humidity of 79%. The conjugates were stored in disposable polypropylene containers at 2 °C until further use.

Conjugate dispersions were prepared by dissolving 0.4 g of freeze-dried solids in an appropriate amount of deionized water, containing 0.02 *w*/*w* % sodium azide and 10 g citrate buffer (50 mM, pH 5), and then stirred at 500 rpm and 25 °C for 1 h on a magnetic stirrer to ensure full hydration, which was confirmed by the absence of visible solids. The buffer was used to ensure the same ionic strength conditions were present as for the conjugates prepared by electrostatic cross-linking and laccase catalysis. The 0.4 *w*/*w* % SBP-SC conjugate via Maillard reaction dispersion was labelled as SBP:SC M dispersion (M to denote Maillard reaction).

### 2.5. Analytical Methods

#### 2.5.1. Conjugate Size

The size of the different conjugates was quantified as the z-average as determined with a Zetasizer (Zetasizer Nano ZS (Malvern Panalytical, Malvern, UK)). Prior to analysis, conjugate samples were diluted with deionised water to a total polymer concentration of 0.01 *w*/*w* %, which gave an optimal sample concentration for measurement, as indicated by the instrument’s software, and then filtered, using syringe filters to remove larger particles, such as dust and other contaminants. 3 mL of sample was then enclosed in a single-use disposable sizing cuvette (DTS0012, Sarstedt, Nümbrecht, Germany) and placed into the equipment at 20 °C. The refractive index and absorption were set to be 1.450, and 0.001, respectively.

#### 2.5.2. Zeta-Potential Measurement

To determine the ζ-potential of the SBP-SC conjugates, a particle electrophoresis instrument (Delsa Nano C, Meritics, High Wycombe, UK) was used at 20 °C. Before the measurement, samples were diluted to 0.2 *w*/*w* % with deionized water. The samples were measured 1 day after preparation.

#### 2.5.3. Shear Viscosity

The shear viscosity of SBP-SC conjugate dispersions prepared at pH 4.5 was analysed, using a rotational rheometer (Physica MCR 301, Anton Paar, Graz, Austria) fitted with a double gap cylinder geometry (DG26.7/T200/SS, Anton Paar, Graz, Austria). Before the measurement, the samples were shaken well in a vial to separate the aggregated conjugates. Then, each sample was loaded into the rheometer measurement cell and allowed to equilibrate at 20 °C for 2 min. Then, the steady shear behaviour of the samples was assessed as a function of shear rate from 1 to 100 s^−1^.

### 2.6. Environmental Stress Tests

The conjugate dispersions were submitted to a pH and a temperature stress test, followed by storing at 25 °C for 24 h before analysis with the same analytical methods, as applied to the freshly prepared samples.

#### 2.6.1. pH Adjustment

The pH stress tests included, adjusting the pH of SBP-SC conjugates with 1 M HCl, 1 mM HCl, and 1 M NaOH, as required, to a value of 4.5 or 7. These two pH values were selected because SBP-SC conjugates might split at both pH 4.5 and pH 7, as a result of residual charges of SC, and opposite charges between two biopolymers, respectively.

#### 2.6.2. Thermal Stress

A thermal stress test was designed as follows. SBP:SC 1:1 P, SBP:SC 1:1 E, and SBP:SC 1:1 M conjugates at pH 4.5 and pH 7 were transferred to a 100 mL vial and sealed with a plastic cap. The vial was incubated in a water bath at 80 °C for 10 min. The samples were stored at 25 °C.

### 2.7. Statistical Analysis

All measured results are reported as the mean (*n* = 3) ± 1 standard deviation of triplicate freshly independent-prepared samples. The data were statistically analysed for significant difference (*p* < 0.05) applying the student’s *t*-test by using Microsoft Excel 2010 (Microsoft, Seattle, WA, USA).

## 3. Results and Discussion

### 3.1. Physico-Chemical Properties of the SBP-SC Conjugates

#### 3.1.1. Visual Appearance

Figure 1 shows test tubes with the three types of SBP-SC conjugates prepared in this study at pH 4.5, pH 7 before and after heat treatment 80 °C for 10 min. At pH 4.5 and before heat treatment, test tubes labelled “1”, all conjugates had sedimented and the supernatant phases were clear except for the P conjugate, which had a milky white supernatant phase. Precipitate colour was white for P and M conjugates while the E conjugate was brown (appearing as grey in the b/w image). It can, therefore, be postulated that the microstructure of these conjugates differed with conjugate type. The appearance of the test tubes, labelled “3” in Figure 1, reveals that all three types of conjugates became transparent following adjustment of their pH to pH 7. This indicates the absence of large aggregates.

However, there were no obvious differences in visual appearance between unheated and heated SBP-SC conjugates, as revealed by comparing test tubes 1 with test tubes 2, and test tubes 3 with test tubes 4. This observation confirms previous report that polysaccharides may protect proteins from aggregation during heat treatment [21].

#### 3.1.2. Conjugate Size and Zeta Potential

##### Prior to Heat Treatment

In order to analyse the particle size of the SBP-SC conjugates, the z-average radius was determined by dynamic light scattering (DLS). The results are shown in Figure 2. At pH 4.5, P conjugates were characterised by a larger particle size than seen in E conjugates (*p*-value < 0.05). The particle size of the M conjugates at pH 4.5 is not reported, as the dispersion was fully precipitated once preparation was complete, and thus the particle size was out of the measurement range of the equipment, i.e., larger than 2,000 nm. When the pH conditions of the dispersions were adjusted from pH 4.5 to pH 7, the particle size of the P conjugate decreased to from (259 ± 23) nm to (183 ± 31) nm, which was similar to that observed for the E conjugate (188 ± 19) nm at pH 7 (*p*-value > 0.05). The particle size of the E conjugate did not change significantly on pH adjustment (*p*-value > 0.05). The particle size of the M conjugate (213 ± 19) nm was larger than the particle sizes for the P and E conjugates at pH 7 (*p*-value < 0.05). When the P and M conjugates were at pH 4.5, the white precipitates were a result of SC self-assembly under acidic conditions. Consequently, both conjugates possessed a large particle size at pH 4.5. Comparatively, the E conjugate displayed a brown precipitate. This result may indicate a lower level of SC aggregation in the E conjugate, leading to a precipitate colour resembling the colour of the SBP solution (brown) and small particle size at pH 4.5. When the P and M conjugates were adjusted from pH 4.5 to pH 7, their particle size decreased. This result corresponds to the visual observation that P and M conjugates showed white precipitates at pH 4.5 that become transparent as the pH was raised to pH 7. The decreased particle size for both conjugates implies that the SC conjugates were de-agglomerated at pH 7. There was no significant change for the particle size of the E conjugate as the pH was adjusted from pH 4.5 to pH 7, although the precipitates of the dispersion were changed slightly under this adjustment. This indicates that less aggregation was observed in the E conjugate dispersion at pH 4.5, suggesting that the E conjugate was more acid-tolerant.

To measure the charge of the SBP-SC conjugate dispersions, the ζ-potential was determined. The results are shown in Figure 3. Among these, the ζ-potential of the P conjugate at pH 4.5 was the most negative of the three conjugate types while the E conjugate had the same negative ζ-potential as the M conjugate. Although the ζ-potential of SC at pH 4.5 was not tested in this study, it was reported in a previous study that SC polymers possess a small positive charge at pH 4.5 [10], which is just below the isoelectric point (pH ~4.6) of SC. Consequently, the negative ζ-potential of the SBP-SC conjugates is likely to be the result of the presence of SBP polymers creating a net negative charge. In addition, both, the E and M conjugates possessed lower net charges at pH 4.5 than the P conjugate, which implies that there were buried negative carboxyl groups within the E and M conjugates. Although the preparation of the E conjugates was based on the P conjugates, the addition of laccase would have altered the structure of the conjugates. These buried carboxyl groups in the E conjugate may thus be the result of covalent cross-links between the ferulic acid present in the SBP and certain amino acids, such as tryptophan, cysteine and tyrosine in the SC, during laccase catalysis [21]. The buried negative groups in the M conjugates were the result of the reducing end of the carbohydrate cross-linking with the amino acids present in the protein during the Maillard reaction. Although the ζ-potentials of the E and M conjugates were similar and especially negative (at both pH values), the visual observations and the particle size data revealed precipitation and aggregation of the M conjugates at pH 4.5. This suggests that the protein moieties of the Maillard conjugates interacted hydrophobically, as charge repulsion between these moieties would have been negligible, due to the pH being close to the IEP of the protein. It appears that the net conjugate charge was dominated by the properties of the SPB. When the pH of the SBP-SC conjugates was adjusted to pH 7, a more negative ζ-potential was observed for all three SBP-SC conjugates, which may be the result of the negative charge of SC at pH 7. In addition, there were no significant differences in ζ-potential between the P, E and M conjugates at pH 7 (*p*-value > 0.05).

##### Post Heat Treatment

To assess thermal stability, an important factor for the application of food ingredients in general, and particularly for those with structure functionality, the conjugate dispersions were heat treated at 80 °C for 10 min. Firstly, particle size data (Figure 2) revealed that the particle size of the P conjugate decreased as a result of the heat treatment at both pH 4.5 and pH 7 (*p*-value < 0.05). In addition, there was no significant change (*p*-value > 0.05) in the ζ-potential of the P conjugate after heat treatment at pH 4.5, whereas it became more negative at pH 7. A decrease particle size of the P conjugate at pH 4.5 after heat treatment suggests that a structural rearrangement took place during heat treatment. Heat induced structural rearrangement of protein-polysaccharide conjugates from a random structure, to a particle characterised by a protein core with a surrounding polysaccharide shell, has previously been discussed [34,40,41,42,43,44]. The protein core formation could be related to the denaturation of casein at pH 4.5 at 80 °C [45,46]. Smaller particle size and higher ζ-potential may suggest that a structural change of the P conjugate at pH 7 during heat treatment. Due to the strong electrostatic repulsion between SBP and SC biopolymers under neutral conditions, heat treatment can lead to the separation of a fraction of the protein from the conjugates increasing the net negative charge of the system [44]. Secondly, heating had a lesser effect on the E conjugate, as indicated by the particle size remaining constant, but the ζ-potential decreased slightly at pH 4.5. This change can be attributed to the heat-induced weakening of the unfolded structures of SBP-SBP conjugates in E conjugates, because the covalent bond weakened after heat treatment, leading to the exposure of some negatively charged groups. The fact that the ζ-potential of the E conjugates held at pH 7 was not affected by heating suggests that separation of the E conjugate was either completely absent, due to the molecularly cross-linked nature of this conjugate type, or too insignificant to affect the ζ-potential. Finally, the consequence of heat treating the M conjugate type held at pH 4.5 could only be assessed by the ζ-potential, as the particles were still too large for the selected particle sizing method (DLS). A less negative ζ-potential was recorded for the M conjugate after heat treatment at pH 4.5, increasing from (−29.18 ± 0.52) mV to (−28.26 ± 0.24) mV, shown in Figure 3. This result was probably due to the structural rearrangement of SBP-SC conjugates contributing to reducing the negative charge. The results at pH 7 revealed that the particle size of the M conjugate decreased from (213 ± 19) nm to (160 ± 14) nm after heating (*p*-value < 0.05), shown in Figure 2. In addition, no significant change in ζ-potential at pH 7 after thermal treatment (*p*-value > 0.05) was found, see Figure 3. The smaller particle size may suggest that the M conjugates were slightly less aggregated after heat treatment.

#### 3.1.3. Steady Shear Viscosity

The conjugate dispersions were assessed for steady shear viscosity behaviour and the results are shown in Figure 4. At pH 4.5 and prior to heating, but in fact, after heating as discussed later, all three types of conjugate dispersions were shear-thinning. This shear-thinning behaviour was more pronounced for the M conjugate dispersion, compared to the P conjugate dispersion, which was more shear thinning than the E conjugate dispersion, see Figure 4a. The more pronounced shear-thinning of the M conjugate dispersion would have been the consequence of the aggregated state of the conjugate [47]. Analogously, as the P conjugate demonstrated stronger aggregation than the E conjugates, their degree of shear-thinning was comparatively higher. One of the most important parameters affecting the rheology of dispersed systems is particle volume fraction, and shear-thinning behaviours are the result of intermediate volume fractions (0.1 < ϕ < 0.5) [48]. It has been reported that higher particle volume fractions in this intermediate volume range result in higher shear viscosity because of the formation of chains and networks of interacting particles. In this case, the magnitude of the shear viscosity is related to the particle size and the coefficient of friction between particles [49]. In this study, the larger aggregated particles of the M conjugates render this conjugate dispersion the highest viscous of the three conjugate dispersions. When the pH condition was adjusted to pH 7, see Figure 4b, shear-thinning was less pronounced for P and M conjugates. This result further evidences the aggregation of P and M conjugates at pH 4.5 but not at pH 7. Comparatively, the E conjugates were less influenced by the altered pH condition, with no change in flow behaviour as the conditions changed from pH 4.5 to pH 7.

To understand the viscosity behaviours of the SBP-SC conjugates after thermal treatment, rotational rheological measurement was performed on the heated conjugate dispersions. In Figure 4c, it is revealed that there was a decrease in viscosity for both P and M conjugates, as compared with unheated SBP-SC conjugates at pH 4.5. The decreased particle size may contribute to the decrease of the shear viscosity for the P conjugate dispersion after heat treatment as it means a lower particle volume fraction. The decreased viscosity of the M conjugate dispersion after heat treatment suggests that the particle size of the conjugates was lower after heat treatment (but still >2000 nm, as explained earlier). As previously interpreted from the ζ-potential results, following heat treatment at pH 4.5, the M conjugates most likely had a core of denatured SC, surrounded by a shell of SBP. Comparatively, a slight decrease and no obvious change in viscosity were observed for the P and E conjugate dispersions, as a result of heating at pH 4.5, comparison data is shown in Figure 4a,c. This phenomenon may be a result of the decreased particle size of the P conjugates due to heating and the heat had less effect on the size of the E conjugates, see Figure 2, heating at pH 7 on the other hand did not change viscosity for any of the three dispersions, comparison data is shown in Figure 4b,d. On analysing the particle size and ζ-potential, there were no significant changes for the E conjugate at pH 7 after heat treatment, and thus no shear viscosity changes. This result further shows the reduced effect of heating on the E conjugate. However, decreased particle sizes were observed for both P and M conjugates, which may contribute to less viscosity in both dispersions after heat treatment. In the shear viscosity results, such phenomena were not observed, perhaps as the particle size decrease was too little.

### 3.2. Microstructure Model

The physico-chemical data, acquired on the three types of SBP-SC conjugates, were interpreted in terms microstructure models pre- and post-heating, see Figure 5a,b respectively. The references in the figure captions refer to previously published models for protein-polysaccharide conjugate structures, but non SBP-SC.

A possible structure of the SBP-SC conjugates at pH 4.5 and pH 7 is shown in Figure 5a. P and E conjugates were hypothesised to possess similar structures at pH 4.5 because of the folded structure of the SC biopolymer at this for casein proteins unfavourable pH. However, there were slight differences because of the additional laccase catalysed cross-linking between SBP and SBP within E conjugate, leading to a higher branched structure of SBP in the E conjugate [26]. In contrast, there is a high degree of covalent bonding between the amine groups of the SC and the aldehydes groups of the SBP during the Maillard reaction, leading to unfolded SC and SBP structures within M conjugates. Under acid conditions, the unfolded SC in the M conjugates is aggregated, resulting in more pronounced aggregation and thus precipitation. At pH 7, similar aggregation properties were observed for P, E, and M conjugates, favouring the coil confirmation of SBP and SC, and suggesting that a similar structure pertains among all three SBP-SC conjugates. Moreover, the large particle size for M conjugate may be a result of more than one SBP cross-linked with SC during Maillard reaction [50]. Comparatively, only one the SBP molecule, cross-linked with one SC molecule in both P and E conjugates [51]. A possible structural rearrangement of P, E and M conjugates after heat treatment at pH 4.5 is shown in Figure 5b. Previous studies suggested that the structural rearrangement of SBP-SC conjugates, during heat treatment, may reinforce the stability of SBP-SC conjugates because of the structure of a protein core with a surrounding polysaccharide shell [44].

## 4. Conclusions

In this study, the aggregation properties of electrostatically-stabilised, laccase-catalysed and Maillard cross-linked SBP-SC conjugates were studied at different environmental stresses. The pH and thermal treatment affected the aggregation properties of all three types of conjugates. The E conjugate was most acid tolerant, followed by the P conjugate, while the M conjugate was the least acid-tolerant. This conclusion is based on the lowest precipitate volume, smallest mean particle size and least degree of shear-thinning behaviour of this conjugate type at pH 4.5. When the condition was adjusted to pH 7, all three dispersions revealed similar aggregation properties. Based on the aggregate property data, it could be concluded that the P conjugate was the structure of folded SC cross-linked with SBP, E conjugates was the structure of folded SC, cross-linked with more compact, higher branches SBP, and M conjugates was the structure of unfolded SC cross-linked with one or two SBPs. Heat treatment led to a structuring re-arrangement with the degree of re-arrangement depending on conjugate type. At pH 4.5 the microstructure model proposes a protein core with a surrounding polysaccharide shell. During heating at pH 7, SC separated from P conjugates, whereas E and M conjugates were hardly affected by heating, probably as SC and SBP were covalently cross-linked. So, it can be concluded that the covalently cross-linked conjugates were more heat-resistance than the electrostatically cross-linked conjugate and would, therefore, be a preferred choice for acidic emulsion-based food and drink formulations requiring pasteurisation.

## Figures and Tables

**Figure 1 foods-08-00192-f001:**
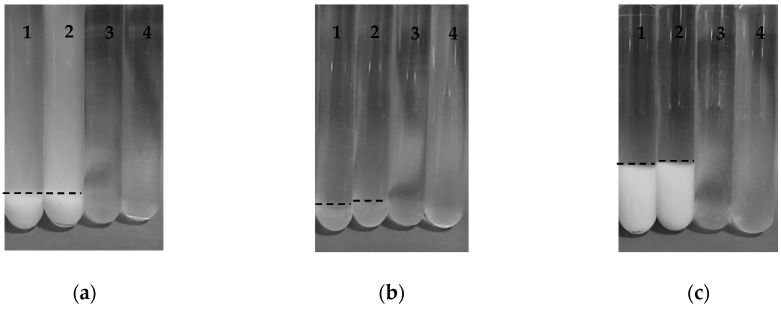
Images of (**a**) SBP-SC P conjugates, (**b**) SBP-SC E conjugates, and (**c**) SBP-SC M conjugates dispersions (1) at pH 4.5; (2) at pH 4.5 heated at 80 °C for 10 min; (3) at pH 7; and (4) at pH 7 heated at 80 °C for 10 min, respectively. The dotted line is the phase boundary of the sediment.

**Figure 2 foods-08-00192-f002:**
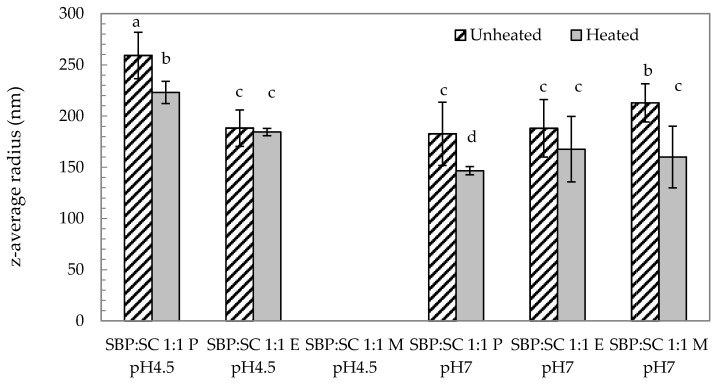
Particle size (z-average radius) of unheated (patterned bars) and heated (filled bars; 80 °C for 10 min) of SBP-SC conjugates at pH 4.5 and pH 7, analysed at 20 °C. The letters (**a**–**d**) represent significant differences among samples (*p*-value < 0.05). No data are shown for SBP:SC 1:1 M, unheated and heated, as the values were outside the measurement range (upper limit of 2000 nm).

**Figure 3 foods-08-00192-f003:**
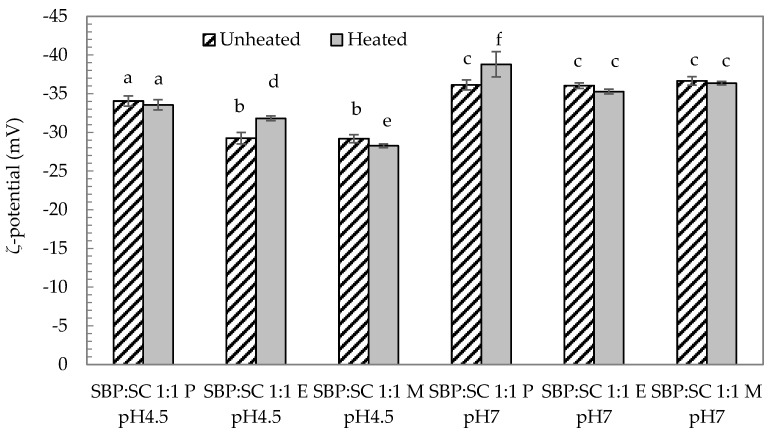
ζ-potential of unheated (upward) and heated (filled) at 80 °C for 10 min of SBP-SC conjugates at pH 4.5 and pH 7 at 20 °C. The different letters (**a**–**f**) represent significant differences among samples (*p*-value < 0.05).

**Figure 4 foods-08-00192-f004:**
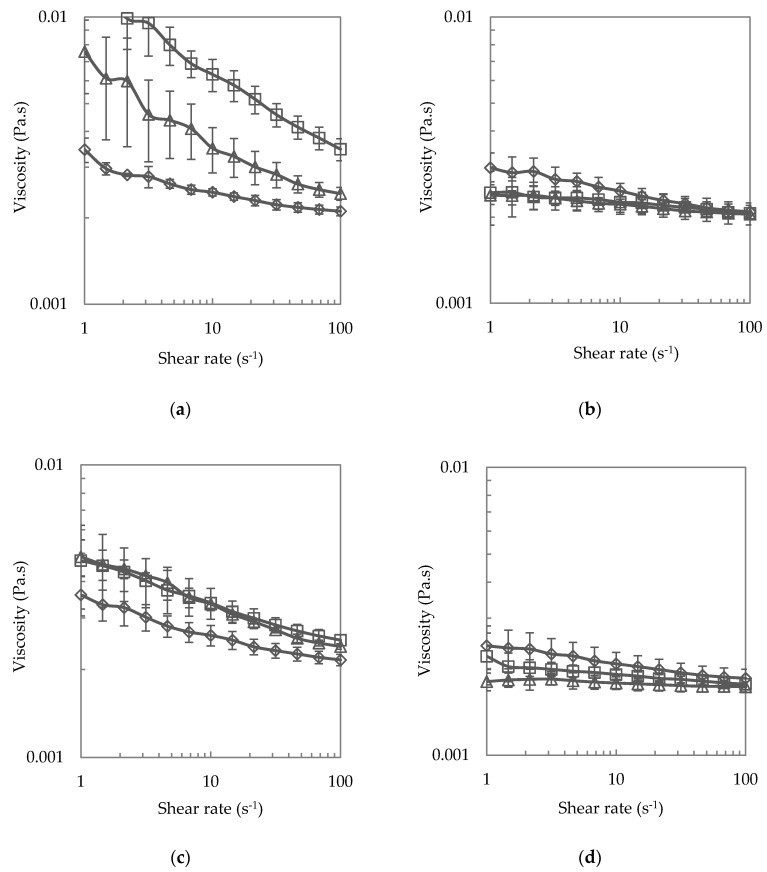
Steady shear viscosity (**a**) at pH 4.5; (**b**) at pH 7; (**c**) heated at 80 °C and pH 4.5 for 10 min; (**d**) heated at 80 °C and pH 7 for 10 min of SBP:SC 1:1 P (triangle), SBP:SC 1:1 E (diamond), and SBP:SC 1:1 M (square) conjugates measured at 20 °C. The values are means, and the error bars correspond to a ± 1 standard deviation of the triplicate measurements taken from freshly prepared samples.

**Figure 5 foods-08-00192-f005:**
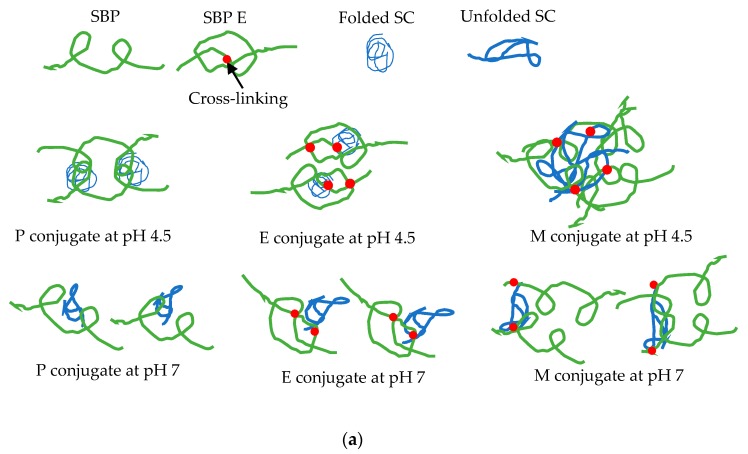
Schematic of (**a**) SBP-SC conjugate via complexation [2], laccase catalysis [30], and Maillard reaction [6,31] at pH 4.5 and pH 7; and (**b**) the structure of P and M conjugates at pH 4.5 heated at 80 °C for 10 min [38].

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
