# Peer review of "Physico-Chemical Properties of Sugar Beet Pectin-Sodium Caseinate Conjugates via Different Interaction Mechanisms"

_foods, 2019, doi:10.3390/foods8060192_

Reviewer 1 Report

The manuscript presents some novel recipe for the preparation of a polysaccharide-protein mixture. However, it needs some clarifications and corrections. Here, my detailed remarks:

i)                 The title seems a bit a weird to me because it starts with “effect of their interaction…” and it specifies only later who is the subject.

ii)                In the abstract it is written “it may be desired to choose one set of conjugate partners for various applications.”. The sentence sounds vague to me. Please rephrase/specify what you intend

iii)              In the abstract “ the aggregate properties”. Which properties? Please specify

iv)              Abstract “rheological assessment”. This sounds vague to me. Please specify what you meant

v)                Line 43 “benefits in terms of human health” sounds vague to me. Please specify

vi)              Paragraph 2.5.3. Besides the given info, it should be mentioned for how long a specific shear rate has been kept constant before moving to the next one. These materials area likely thixotropic and their viscosity over time might change even if the shear rate is fixed.

vii)             Also, these materials look viscoelastic. The elastic and viscous moduli should have been measured. Moreover, I wonder that given the high viscosity reported, and their apparent viscoelasticity, it is difficult to properly load the material into a double gap geometry. Plus, wall slip it is likely occurring. So I wonder that it would have been more useful to measure the viscosity with a rough parallel plates or rough cone-plate geometry.

viii)           A picture of the microstructure of the samples, either with an optical microscope or a confocal one might  have been useful to better characterize it.

ix) in the conclusions first line "P, E, M conjugates" are mentioned. I think it would be useful to specify what are these acronyms.

x) always in the conclusions and in another section of the manuscript it is mentioned "to understand the emulsification ability of the three conjugates". But I think that no emulsification was conducted throughout the manuscript. Otherwise stability, droplet size distributions and volume fraction, interfacial tension and the nature of the oil phase should have been reported. So please remove any reference to emulsification or provide the above-mentioned data.

Author Response

We thank the reviewer for their constructive comments and suggestions to improve the quality of our manuscript.

Point 1: The title seems a bit a weird to me because it starts with “effect of their interaction…” and it specifies only later who is the subject.

 Response 1:We have changed the title to “Physico-chemical properties of sugar beet pectin-sodium caseinate conjugates via different interaction mechanisms”

 Point 2: In the abstract it is written “it may be desired to choose one set of conjugate partners for various applications.”. The sentence sounds vague to me. Please rephrase/specify what you intent

Response 2:We changed  the sentence to “The conjugate partners and their interaction type affect performance at acidic or neutral pH and during thermal processing, thus requiring careful selection.”

Point 3: In the abstract “the aggregate properties”. Which properties? Please specify

Response 3:Thanks for this hint. We added “(particle size, conjugate charge, shear viscosity)”

Point 4: Abstract “rheological assessment”. This sounds vague to me. Please specify what you meant

Response 4:We changed it to “shear viscosity”.

Point 5: Line 43 “benefits in terms of human health” sounds vague to me. Please specifc

Response 5:We changed the sentence to “provides essential amino acids.”

Point 6: Paragraph 2.5.3. Besides the given info, it should be mentioned for how long a specific shear rate has been kept constant before moving to the next one. These materials are likely thixotropic and their viscosity over time might change even if the shear rate is fixed.

Response 6:We used the “record steady state data” option of the rheometer software. The dispersion are of quite low viscosity and we did not expect to see thixotropy, such as a hysteresis loop in up and down shear measurements. 

Point 7: Also, these materials look viscoelastic. The elastic and viscous moduli should have been measured. Moreover, I wonder that given the high viscosity reported, and their apparent viscoelasticity, it is difficult to properly load the material into a double gap geometry. Plus, wall slip it is likely occurring. So I wonder that it would have been more useful to measure the viscosity with a rough parallel plates or rough cone-plate geometry.

Response 7:Thank you for these observations and suggestions. In hindsight this may have indeed been useful although oscillatory shear measurements would potentially have been a challenge due to the low viscosity of the dispersions. We have instead focussed our research on assessing the performance of these conjugates at emulsion interfaces. A manuscript is in preparation.

Point 8: A picture of the microstructure of the samples, either with an optical microscope or a confocal one might  have been useful to better characterize it.

Response 8: This is a valid comment, but unfortunately, we do not have micrographs of the samples. We later imaged emulsions stabilised with these conjugates.

Point 9:  in the conclusions first line "P, E, M conjugates" are mentioned. I think it would be useful to specify what are these acronyms

Response 9:Thanks for this hint, we have followed the advice.

Point 10: always in the conclusions and in another section of the manuscript it is mentioned "to understand the emulsification ability of the three conjugates". But I think that no emulsification was conducted throughout the manuscript. Otherwise stability, droplet size distributions and volume fraction, interfacial tension and the nature of the oil phase should have been reported. So please remove any reference to emulsification or provide the above-mentioned data.

Response 10:Thanks for this valid comment. We have followed the advice.

 Reviewer 2 Report

The manuscript deals the aggregation properties of sugar beet pectin -sodium caseinate conjugates at acidic and neutral pH as well as their thermal processing stability. The target of the manuscript was to obtain food structuring ingredients, for the application as emulsifier. By means of particle size, zeta potential and rheological measurements the authors conclude that that enzymatically cross-linked SBP:SC was more acid tolerant than electrostatically interacting conjugates. The manuscript appears interesting and well written. The authors should may try to amplify how this research work contributes to forwarding the field of study. For these reason in my opinion, the manuscript could be revised according to the following guidelines:

Title: I would suggest to change the title in a more endearing style…Starting with  “Effect of their interaction mechanism… “ sounds strange.

Methods –lines 205-215 DLS and zeta potential experiments. Can the authors provide references that report the dilution procedure? What about the values without dilution (and filtering)?

Fig 2. axis of zeta potential should be corrected value of the sample SBP:SC 1:1 MpH4.5 is out of scale.

Table 1- since table 1 reports the same data showed in Fig 2 and 3 I would suggest to move this table in the supporting and information sections.

-Discussion since the item of the present work was recently afforded from other authors (Effect of the coexistence of sodium caseinate and Tween 20 as stabilizers of food emulsions at acidic. DOI: 10.1016/j.colsurfb.2018.02.003), were it was ascertained that, at pH close to caseinate isoelectric point, emulsions stabilized with the blend of caseinate and Tween 20 were more stable, compared with emulsions stabilized only with sodium caseinate. I would suggest to take into account in the discussion the these recent results.

-I suggest to merge fig 5 and 6 in a single panel.

As a whole the authors could try to further emphasize the relevance of this work to new findings in this field and the ways in which the new results have advanced the field.

Author Response

We thank the reviewer for their constructive comments and suggestions to improve the quality of our manuscript. We are in particular grateful for the literature pointed out.

Point 1: Title: I would suggest to change the title in a more endearing style…Starting with “Effect of their interaction mechanism… “sounds strange.

Response 1:We have changed the title to “Physico-chemical properties of sugar beet pectin-sodium caseinate conjugates via different interaction mechanisms”.

Point 2: Methods –lines 205-215 DLS and zeta potential experiments. Can the authors provide references that report the dilution procedure? What about the values without dilution (and filtering)?

Response 2: We followed the advice by the manufacturer and added the following “which gave the optimal sample concentration for measurement as indicated by the instrument’s software”.

Point 3:Fig 2. axis of zeta potential should be corrected value of the sample SBP:SC 1:1 MpH4.5 is out of scale.

Response 3: We have removed the data from the graph and added the following explanaiotn into the legend:”No data are shown for SBP:SC 1:1 M, unheated and heated, as the values were outside the measurement range (upper limit of 2,000 nm).

Point 4:Table 1- since table 1 reports the same data showed in Fig 2 and 3 I would suggest to move this table in the supporting and information sections.

Response 4:Thank you for this advice, we have created a supporting section.

Point 5:Discussion since the item of the present work was recently afforded from other authors (Effect of the coexistence of sodium caseinate and Tween 20 as stabilizers of food emulsions at acidic. DOI: 10.1016/j.colsurfb.2018.02.003), were it was ascertained that, at pH close to caseinate isoelectric point, emulsions stabilized with the blend of caseinate and Tween 20 were more stable, compared with emulsions stabilized only with sodium caseinate. I would suggest to take into account in the discussion the these recent results.

Response 5:Thank you for pointing out this literature reference. We are currently preparing a separate manuscript on emulsions stabilised (or not) with our conjugate dispersions where we will consider “emulsion” literature. Based on comments by reviewer 1, we feel that we will confuse the reader including emulsion references in the current manuscript.

Point 6:I suggest to merge fig 5 and 6 in a single panel.

Response 6:Thanks for the advice. We have combined the figures.

Point 7:As a whole the authors could try to further emphasize the relevance of this work to new findings in this field and the ways in which the new results have advanced the field.

Response 7:We added a sentence in line 484 “it can be concluded that the covalently crosslinked conjugates were more heat-resistance than the electrostatically crosslinked conjugate and would therefore be a preferred choice for acidic emulsion-based food and drink formulations requiring pasteurisation.”

Round  2

Reviewer 1 Report

The authors addressed all my remarks. I recommend the paper to be published as is.

Reviewer 2 Report

The authors addressed almost  my suggestions. The manuscript in the present form appears  improved thus I recommend the acceptance.